# West Nile Virus and Tick-Borne Encephalitis Virus Are Endemic in Equids in Eastern Austria

**DOI:** 10.3390/v13091873

**Published:** 2021-09-19

**Authors:** Phebe de Heus, Jolanta Kolodziejek, Zdenĕk Hubálek, Katharina Dimmel, Victoria Racher, Norbert Nowotny, Jessika-M. V. Cavalleri

**Affiliations:** 1Clinical Unit of Equine Internal Medicine, University of Veterinary Medicine Vienna, Veterinärplatz 1, 1210 Vienna, Austria; Phebe.de-Heus@vetmeduni.ac.at (P.d.H.); Jessika.Cavalleri@vetmeduni.ac.at (J.-M.V.C.); 2Viral Zoonoses, Emerging and Vector-Borne Infections Group, Institute of Virology, University of Veterinary Medicine Vienna, Veterinärplatz 1, 1210 Vienna, Austria; Jolanta.Kolodziejek@vetmeduni.ac.at (J.K.); Katharina.Dimmel@vetmeduni.ac.at (K.D.); 3Institute for Vertebrate Biology, Czech Academy of Sciences, Kvĕtná 8, 60365 Brno, Czech Republic; zhubalek@brno.cas.cz; 4Department of Mathematics, University of Salzburg, Hellbrunner Straße 34, 5020 Salzburg, Austria; victoria.racher@stud.sbg.ac.at; 5Department of Basic Medical Sciences, College of Medicine, Mohammed Bin Rashid University of Medicine and Health Sciences, Dubai Healthcare City, Building 14, Dubai P.O. Box 505055, United Arab Emirates

**Keywords:** flavivirus, West Nile virus, tick-borne encephalitis virus, Usutu virus, seroprevalence, horses, epidemiology, Austria

## Abstract

The emergence of West Nile virus (WNV) and Usutu virus (USUV) in addition to the autochthonous tick-borne encephalitis virus (TBEV) in Europe causes rising concern for public and animal health. The first equine case of West Nile neuroinvasive disease in Austria was diagnosed in 2016. As a consequence, a cross-sectional seroprevalence study was conducted in 2017, including 348 equids from eastern Austria. Serum samples reactive by ELISA for either flavivirus immunoglobulin G or M were further analyzed with the plaque reduction neutralization test (PRNT-80) to identify the specific etiologic agent. Neutralizing antibody prevalences excluding vaccinated equids were found to be 5.3% for WNV, 15.5% for TBEV, 0% for USUV, and 1.2% for WNV from autochthonous origin. Additionally, reverse transcription quantitative polymerase chain reaction (RT-qPCR) was performed to detect WNV nucleic acid in horse sera and was found to be negative in all cases. Risk factor analysis did not identify any factors significantly associated with seropositivity.

## 1. Introduction

The increasing emergence of vector-borne viruses in Europe causes rising concern for public and animal health [1,2,3,4,5,6,7,8]. The genus *Flavivirus* of the family *Flaviviridae* incorporates important human and animal arthropod-borne pathogens such as West Nile virus (WNV), Usutu virus (USUV), and tick-borne encephalitis virus (TBEV). All three viruses have been associated with neurologic signs and have been known to circulate in Austria [9,10]. Incidences of West Nile disease (WND), USUV infections, and tick-borne encephalitis (TBE) fluctuate extensively on a yearly basis, depending on factors that influence vectors, hosts, transmission route, and viral replication [10,11,12,13]. The transmission cycles of WNV and USUV are very similar with birds acting as a reservoir and amplifying host, and mosquitoes (mainly *Culex pipiens*) being the principal vector [6,8,14,15,16,17].

West Nile virus lineage 2 has been reported in Austria since 2008 in several species including birds of prey, humans, and mosquitoes [3,8,9,13,16,18,19]. For WNV, horses and humans represent dead-end hosts. The first equine case of West Nile neuroinvasive disease (WNND) in Austria was diagnosed in 2016 at the University of Veterinary Medicine of Vienna [16,20]. Of the eleven confirmed WNND cases that were treated at the Equine University Hospital between 2016 and March 2021, two presented in 2016, three in 2017, two in 2018, three in 2019, and one in 2020. All cases showed gait abnormalities and the majority displayed muscle fasciculations and a change in mentation. Six horses were euthanized because of the severity of clinical signs [20].

In 2001, USUV was detected for the first time on the European continent, more specifically in Austria, and was associated with bird mortality [1,21]. Retrospectively, it was identified in earlier cases of blackbird die-off in Northern Italy in 1996 [2]. The virus was detected in native mosquito pools (*Cx pipiens*/*Cx torrentium*) [8,13,14,15,16,17] as well as in an invasive mosquito species (*Aedes japonicus japonicus)* [17]. Human infections are mostly asymptomatic, but can induce rash or, rarely, neuroinvasive disease, particularly in immunocompromised patients [22,23,24]. USUV RNA was identified in human blood donations from three eastern Austrian federal states in 2017 with an increasing number of infections in 2018 in southern and western federal states [9,25]. None of the donors reported clinical signs except for a single donor reporting a skin rash. Neutralizing antibodies against USUV have been found in blood from healthy horses in Croatia and Poland [26,27]. Serologic evidence of USUV infection was also reported for horses in Italy [28]. The occurrence of USUV in horses in Austria and potential clinical implications are presently unknown.

TBEV is primarily transmitted by hard ticks, specifically *Ixodidae*, with small mammals representing the main reservoir hosts [29]. Horses and humans can be infected, but similar to WNV are dead-end hosts. Historically, high TBEV-associated morbidity rates among humans in Austria triggered monitoring and mass vaccination programs over decades [10,30]. Despite this, no confirmed equine clinical cases of TBE were diagnosed at the University of Veterinary Medicine Vienna during the last 20 years.

In 2011, a serologic study focusing on flavivirus infections in horses in Austria found a seroprevalence of 26.1% for TBEV [31]. The authors, however, detected neither WNV nor USUV neutralizing antibodies in their study population.

The prevalence of WNV and of USUV amongst the horse population in Austria is presently unknown. Hence, this study aimed to determine the seroprevalence of WNV, USUV, and TBEV in equids in eastern Austria and determine risk factors for seropositivity. The presence of WNV nucleic acids was also investigated.

## 2. Materials and Methods

### 2.1. Study Design and Population

A prospective cross-sectional prevalence study was conducted. Sampling of blood from equids took place at the peak of the mosquito season, between July and October, 2017. The samples were conveniently collected from horses and donkeys of at least one year of age from patients hospitalized at the University of Veterinary Medicine Vienna and from privately owned, clinically unremarkable equids recruited in eastern Austria. The three cases with a confirmed diagnosis of acute WNND in 2017 were excluded from the study.

This study was approved by the Ethics Committee of the University of Veterinary Medicine Vienna, the Austrian Federal Ministry of Labour, Social Affairs, Health and Consumer Protection, and the Austrian Federal Ministry of Education, Science and Research under the Austrian Federal animal use license BMWF-68.205/0125-WF/V/3b/2017. Informed consent was obtained for sampling and for the use of data.

### 2.2. Questionnaire and Data Collection

Owners or, if not accessible, caretakers of all study animals completed a questionnaire regarding vaccination against WNV, disease history, transport history, stable type (i.e., box or pasture), stable location, length of stay in the yard, open water sources in the surroundings of the horse, and the use of insect repellent measures (chemical and physical) (Appendix A). For all animals, the university hospital records were searched for existing veterinary records. Horse identification documents were checked for demographic data and vaccination records. For hospitalized equids, the primary complaint according to the hospital records was documented.

A Google Maps (Google LLC, Mountain View, CA, USA) search of the stable surroundings for location of natural water sources and national parks was conducted for each of the premises. The federal state was derived from the stable postal code. Lower Austria was analyzed as separate eastern and western parts, based on the first digit of the postal code, in order to detect a possible prevalence trend in a geographic east–west direction.

In addition to inquiring as to the transportation history from the owner/caretakers, transport and import to Austria was cross-referenced with the animal’s passport (i.e., location of vaccination). In horses with serologic evidence of WNV exposure, the exact birthplace was investigated.

Equids sampled in the field (conv) were subjected to a clinical examination and venous blood withdrawal from the jugular vein. Venous blood for the study purpose in hospitalized equids (hosp) was taken as part of the diagnostic process or prior to sedation in order to avoid unnecessary venipuncture. Clinical examination data in hosp equids were extracted from the hospital records. After collection, blood samples were subsequently centrifuged and serum samples were separated, aliquoted, and stored between −20 and −80 °C until analysis.

### 2.3. Flavivirus Antibody Detection in Serum Samples

IgG and IgM ELISAs, both detecting antibodies against the flavivirus structural envelope protein (pr-E), were performed with commercial kits according to manufacturer’s instructions (ID Screen^®^ Competition Multi-species ELISA Kit and ID Screen^®^ West Nile IgM Capture ELISA Kit; both IDvet, Grabels, France). Both kits have been validated for use on equine sera. Van Maanen et al. [32] reported for the IgG competition ELISA a sensitivity of 98% with a 95% confidence interval of 92–100%, and a specificity of 100% with a 95% confidence interval of 98–100%. According to the manufacturers, the IgM capture ELISA showed a specificity of 100% with a 95% confidence interval of 99.31–100% when horses from a disease-free region were tested. Neither a sensitivity percentage nor a confidence interval is mentioned for the IgM ELISA, but results of the validation report identifying all positive samples as positive and a 100% agreement with a national reference laboratory in-house MAC ELISA translate to a 100% sensitivity. ELISA-reactive samples were categorized as either positive, indeterminate, or negative, according to the formulas provided in the manufacturer’s instructions. Samples with positive or indeterminate ELISA results were subsequently analyzed for specific neutralizing antibodies against three flaviviruses (WNV, USUV, and TBEV), utilizing a neutralization microtest, namely the 80% plaque reduction neutralization test (PRNT-80), as described previously [33]. Briefly, two-fold heat-inactivated serum dilutions were tested against 50–100 tissue culture infectious doses (plaque-forming units) of virus. After three to four days, serum neutralization titers (SNT) were determined by examining the cytopathic effect in Vero cell culture. PRNT was considered positive if the plaque count reduction was at least 80%. Titers were recorded as the reciprocal of highest serum dilution that produced at least 80% plaque reduction.

### 2.4. WNV RNA Detection in Serum Samples

To detect WNV nucleic acid, a screening RT-qPCR was performed on all equid serum samples, independent of the serologic results. Briefly, a 200 µL aliquot of each serum sample was used to extract viral RNA using QIAamp 96 Virus QIAcube HT Kit (Qiagen, Hilden, Germany), following manufacturer’s instructions as described previously [34]. Subsequent RT-qPCR targeting the highly conserved 5′ non-coding region of WNV lineages 1 and 2 was performed, as described previously by Kolodziejek et al. in 2014 [35].

### 2.5. Data Analysis and Statistical Methods

All questionnaire and equine passport-derived information, as well as laboratory outcome data and observed information on natural water sources, was entered into the spreadsheet program Excel (Microsoft Corporation, Redmond, WA, USA) (Appendix A). For three horses in group hosp and 47 horses in group conv, a passport was not available to the researchers. For information on WNV vaccination in these cases, veterinary history questionnaire answers from the owner were used. If information from the questionnaire and hospital records or equid passports was contradictory (i.e., vaccination or illness), then the data from hospital and passport documentation were prioritized over the questionnaire. Vaccinated horses were included in the risk assessment analysis. A backward variable elimination based on the Akaike Information Criterion (AIC) using a logistic regression model was applied, with the outcome variable IgG (yes, no). For analyzing the second outcome variable PRNT-based probable agent (negative, WNV, TBEV), multinominal logistic regression models were used. Univariable associations were considered descriptively by pairwise plots and contingency tables. All *p*-values were adjusted for multiplicity at the familywise level α = 0.05 using the Bonferroni procedure. Statistical analyses were conducted using R version 3.5.1 (R Core Team 2018) (R Foundation for Statistical Computing, Vienna, Austria).

## 3. Results

### 3.1. Study Design and Population

In total, 348 equids were sampled, including 334 horses and 14 donkeys (Table 1 and Appendix A). The population consisted of 57 hospitalized equids and 291 field sampled, clinically unremarkable equids, including 143 mares, 179 geldings, 25 stallions, and one horse for which sex was not noted (Table 1). The median age was 12.4 years, excluding the four equids for which age was not documented. The home yards of the equids were located in five federal states (Vienna, Burgenland, Lower Austria, Upper Austria, and Styria). For the recruitment of convenience-sampled unremarkable equids, 30 stable yards were contacted by telephone, of which 23 agreed to participate in the study. Per yard, the number of horses ranged from 1 to 32 equids.

Four horses were vaccinated against WNV two to six years prior to the study period, as documented in their equine passport. They all tested seropositive for IgG and had neutralizing WNV antibody titers ≥ 20, as determined by the PRNT-80 assay. However, one vaccinated horse showed an even higher TBEV than WNV neutralizing antibody titer (Appendix A).

### 3.2. Laboratory Results

#### 3.2.1. Flavivirus Antibody Detection in Serum Samples

Of all samples tested, 90/348 equids tested IgG ELISA-positive and 5/348 had an indeterminate reading (Figure 1). ELISA results for anti-WNV-IgM were all negative with the exception of one indeterminate result, which was subsequently determined to be seronegative for all three flaviviruses by PRNTs (Appendix A).

The prevalence of the three flaviviruses (Figure 2) was based on the highest neutralizing antibody titer ≥20 (Appendix A). This definition assumes one probable agent and excludes the possibility of co-infection with multiple flaviviruses. The prevalence found for the three flaviviruses based on a probable agent, excluding the vaccinated horses, was WNV 5.3% (18/342), TBEV 15.5% (53/342), and USUV 0% (Figure 2b).

The one hospitalized donkey, which was clinically unremarkable but accompanied its sick foal, tested positive for anti-flavivirus IgG and TBEV neutralizing antibodies. Of the further 13 donkeys sampled in the field, three sera tested positive for IgG antibodies and all contained TBEV neutralizing antibodies as well. Consequently, a total donkey seroprevalence of 28.6% for TBEV and 23.1% for the field-sampled donkeys was found. The four seropositive donkeys came from four different stables in two federal states (Lower Austria and Styria).

USUV neutralization titers were borderline positive (20) or negative, with the exception of two cytotoxic and therefore unreliable serum samples. In all but one serum sample with USUV titers of 20, a higher titer was found against another flavivirus. This one sample with an USUV titer of 20 showed indeterminate WNV and TBEV titers of <20, thus the reactions of this serum sample were considered unspecific and the sample was regarded negative for the three flaviviruses tested.

#### 3.2.2. Austrian Autochthonous WNV Infections

The one non-vaccinated WNV PRNT-positive hospitalized horse had a Hungarian passport number, suggestive of being bred in Hungary. A birthplace was unfortunately not recorded for this horse. It could not truly be classified as an autochthonous case.

Based on the available data from the questionnaires and the equid identification papers in group conv, four out of 18 horses could be truly identified as having evidence of former WNV infections from autochthonous origin (4/342, 1.2%). These horses were documented to be born in Austria and had no available documentation indicating travel abroad. Of the remaining 14 seropositive equids, one was vaccinated against WNV. Thirteen horses had a questionable birthplace, a foreign passport, a questionable travel history, or were documented to be imported or traveled abroad. Interestingly, 6/14 of these horses had stayed in a WNV endemic region (Hungary) and a further two might have stayed there.

#### 3.2.3. Detection of WNV Nucleic Acid in Equine Serum

No WNV RNA was found in any of the serum samples.

### 3.3. Geographic Distribution

Taking all studied equids together and including the vaccinated horses, thirteen stables distributed over ten postal codes housed WNV seropositive horses, based on the PRNT80 results. Of these stables, two shared one address. Per stable, one to four horses were WNV seropositive. For TBEV, 23 stables distributed over 19 postal codes contributed one to seven seropositive horses per stable (Figure 3). If a postal code was known for an equid, but not an exact stable address, this was interpreted as one stable, if no other horses were located at that postal code. Of two horses in group hosp, both the stable address and postal code were unknown and therefore excluded from the maps in Figure 3a,b.

### 3.4. Risk Factor Analysis

We started with a complete regression model, considering the variables: age, sex, breed, coat color, respiratory rate, heart rate, rectal temperature, illness <12 months duration, chronic illness, federal state of the stable, stable type, water, national park, import, transport, insect protection, and WNV vaccination. The variable national park refers to whether or not a dedicated nature reserve area was located near the stable. None of the selected variables was significant and none was explanatory for IgG status. Additionally, none of the variables selected for the multinominal logistic regression for the outcome of probable agent was significant nor could explain the outcome. After adjustment of multiplicity, all *p*-values were close to one and therefore not reported.

Although non-significant, the relative risk ratios, derived from the multinominal logistic regression model from variables indicating a trend, are reported in Table 2 for descriptive purposes.

## 4. Discussion

In this study, the antibody prevalence against three flaviviruses was investigated in eastern Austria. Prevalence, based on neutralizing antibodies and excluding the possibility of co-infection as well as vaccinated equids, was found to be 5.3% for WNV, 15.5% for TBEV, and 0% for USUV. Certain autochthonous evidence for former WNV infection was documented in four horses (1.2%). Active WNV infection was investigated, but no WNV RNA was found in serum samples of the study population. We were unable to find risk factors for equids related to management, geographic, or demographic characteristics in eastern Austria.

The prevalence of arboviruses varies yearly and the emergence of an epidemic in an endemic area has been related to climatic conditions that can affect vectors, the abundance of vectors, the exposure to vectors, and virus replication [36]. Higher temperatures were identified as a driving force for WNV disease outbreaks, the number of cases, spread, and increasing transmission competence in mosquitoes [36,37,38]. Heavy rainfall and drought are associated with higher incidence rates [36,37]. The Governmental Meteorologic and Geophysical Service Austria (Zentralanstalt für Meteorologie und Geodynamik, ZAMK) communicates in their annual reports that 2017 was the ninth warmest year since the beginning of recording [39]. In addition, particularly eastern Austria was found to be extremely dry and the maximum temperatures during the summer months were recorded in the city of Vienna [39]. These factors seemed to be contributing factors to the study period of 2017 being a successful transmission season. The prediction of WNV outbreaks on the basis of the current environmental, genetic, transmission, and ecological aspects seems to be difficult, as outlined in a review of the largest European WNV outbreak to date in 2018 [11]. More knowledge on the enzootic transmission cycle is necessary to anticipate outbreak situations.

A limitation of our study was the geographic restriction to the region of WNV outbreak activity in the previous years and the methodology of convenience sampling. Horses can function as sentinel species for arboviruses. Therefore, the equid seroprevalence in our investigated, geographically localized population, where neurologic cases have occurred [4,16,20], is of importance to the one health principle. The eastern Austrian federal states were the focus for the recruitment of equids as active WNV circulation in the greater Vienna area and Lower Austria was found in previous years [16]. Several surveillance studies in Austria in 2015–2016 yielded positive findings in birds, mosquitoes, and human blood donations, in addition to the occurrence of four human and two equine clinical cases [16]. The most frequently encountered mosquito genus was *Culex,* followed by *Aedes*. Both species can transmit WNV. Mosquito infection rates were particularly high in pools selectively trapped near sites where clinical cases had occurred [16]. Results from a long-term surveillance program in Vienna between 2017 and 2019 identified *Ixodus ricinus*, a principal vector of TBEV, as the most abundant tick species [40].

In our population, 21 samples had positive readings (titer ≥ 20) in PRNT-80 against more than one of the tested flaviviruses. Cross-reactivity between the antigenically similar flaviviruses is a diagnostic challenge [41,42,43,44]. The PRNT is the most specific serologic test according to the terrestrial manual of the World Organization for Animal Health (OIE). Though ELISAs with high specificity and differentiation capacity between vaccinated and infected individuals were developed [45], the PRNT is considered the gold standard confirmatory test. The PRNT assay can be used to solve ambiguity between flavivirus infections by testing a panel of related flaviviruses in parallel. A disadvantage of this assay is the higher biosafety level (BSL-3) required to handle infective viruses. If a serum reacts in PRNT against several flaviviruses above cut-off values, the terrestrial manual of the OIE considers the flavivirus with an at least fourfold higher antibody titer as the etiologic virus, compared to the other flaviviruses tested [46,47]. In this study, we did not perform an end-point titration. This obstructs full evaluation of exact titer differences and subsequently the decision making on potential co-infections by other flaviviruses. Positive IgG ELISA readings with uncertain PRNT results, as 18 equids in our study, could have been borderline positive ELISA results, but could also indicate exposure to an unidentified flavivirus [46]. Four horses had been vaccinated against WNV two to six years prior to the study and all showed WNV neutralizing antibody titers at or above 20 for WNV. However, one of them also revealed an even stronger reaction in the TBEV PRNT-80, indicating that further investigation regarding the validity of these conclusions is needed. WNV vaccination titers would have been expected to decline significantly by this time. A natural WNV infection (possible with immunologic memory from vaccination) or a natural TBEV infection in the WNV vaccinated horses seem more plausible explanations for the respective flavivirus neutralizing antibody titers.

### 4.1. West Nile Virus

Compared to previous studies in Austria from the years 1987–2000 [48] and 2011 [31], autochthonous infections of WNV have now been identified in the equid population. WNV IgG antibody titers in horses are reported to stay elevated for at least 15 months post infection [49]. Thus, antibodies detected in the study population did not necessarily originate from infections in 2017 only. In WNV-vaccinated horses, neutralizing antibodies were found up to six years after vaccination. It is difficult to prove whether these antibodies were vaccination-induced or from natural infection. The development of a vaccine and corresponding assay with the capacity to differentiate between infected and vaccinated equines would improve the diagnosis of WNND and help future epidemiologic studies to better subcategorize individuals. Autochthonous prevalence for residence or stay in countries bordering the east of Austria was especially relevant, since these countries (e.g., Hungary) have been documented as endemic for WNV [3,50]. The European Centre for Disease Prevention and Control (ECDC) recorded 204 human and 127 equines cases of WNND during the transmission season of 2017 in the European Union [51]. In Austria’s neighboring country, Slovakia, the seropositivity rate of non-vaccinated horses for WNV was 8.3%, and autochthonous infections with WNV occurred in at least 4.8% of the animals [52]. In Croatia, the reported WNV IgG prevalence in horses of 11.9% [53] was lower compared to the 25.9% IgG prevalence in Austria. However, no mention was made in the Croatian study whether confirming neutralizing assays were performed to identify the exact flaviviral origin of the antibodies. In a study in the northern part of Serbia on 252 horses, 28.6% showed WNV antibodies by ELISA, and all ELISA-positive samples were confirmed by PRNT-90 with neutralizing antibody titers of 23 to >512 [54]. Hungary was the first central European country where a WNV lineage 2 emerged in or just before 2004 [3]. From there, within a few years, the virus spread to neighboring countries as well as eastern, southern, and central Europe [5,18,19,55]. An epizootic involving wild birds infected with WNV lineage 2 as well as horses occurred in 2018 in Germany [56], indicating the rapid west- and northward spread of this strain. In 2018, a remarkable increase in human and equid cases in Europe was reported by the ECDC. Apart from Austria, ten other European countries have reported equine WNND cases (Bulgaria, France, Germany, Greece, Hungary, Italy, Portugal, Slovenia, Romania, and Spain) [57]. As viremia in WNV infections is short and low, it was not surprising that WNV RNA was not found in any animal; nonetheless, it was worth performing this exercise especially on hospitalized horses with unclear diagnosis (e.g., fever bout), where IgM antibodies might not have been detectable at the time of sampling.

### 4.2. Tick-Borne Encephalitis Virus

The current (2017) prevalence for TBEV in Austrian horses (15.6%) falls well within the former Austrian studies of 2011 and 1999 [31,58]. The prevalence study conducted in 2011 in Styria, Lower Austria, and Vienna found neutralization flavivirus antibodies in 26.1% of a horse population, all confirmed to be against TBEV [31]. This was comparable to a small targeted study involving an outbreak in Germany, which found prevalence rates of 20–30% [59], and higher than the reported 2.9% seroprevalence from the German endemic region Marburg-Biedenkopf [60]. An epidemiologic study in Austria´s neighboring country, Slovakia, between 2008 and 2011 found no TBEV neutralizing antibodies, but a further study in 2013 resulted in a 3.4% seroprevalence against TBEV [52,61]. Differences in geographic location, climatic conditions, and sampling of a single breed in the 2011 Austrian study could have influenced these results. TBE has been endemic in certain areas of Austria for decades [10,30]. In eastern Austria, TBE endemic foci date back to the 1970s and even before [10,30]. In 2017, a total of 123 human cases of TBE were recorded in Austria [62], while no equine cases were diagnosed at the University Equine Hospital during that year. It is noteworthy to mention that TBEV infection in horses remains usually subclinical.

### 4.3. Usutu Virus

Reports on USUV in horses are scarce. Rushton et al. detected no neutralizing antibodies against USUV in Austria in 2011 [31], and neither did Hubálek et al. (2013) nor Csank et al. (2018) in the Czech Republic or Slovakia [52,58]. On Mallorca (Spain) and in Croatia, seroprevalence rates of 1.2% and 0.14%, respectively, were reported [26,63]. In Serbia, one of 349 horses sampled in 2009–2010 had cross neutralizing antibodies against USUV and WNV [64]. USUV seroprevalence detected in horses in Poland was 27.98% in 2012/2013, with no reported clinical disease [27]. This number is surprisingly high and could possibly include nonspecific reactions. Savini et al., 2011, reported seroconversion in sentinel horses for USUV, suggesting them as a good monitoring species [28]. Since in our study no sole high titer or unequivocal titer difference to other flaviviruses was found for USUV, no definite proof of USUV infection in equids in Austria was obtained. In 2017, seven USUV-positive human blood donations were detected in the Austrian federal states of Vienna, Lower Austria, and Burgenland [25]. Future studies on USUV in equids should aim to correlate clinical signs to USUV infections, focusing on neurologic signs as this virus is neuroinvasive in certain animal species. Experimental infection studies in equids could further shed light on the role of USUV in equids, especially on the clinical relevance for this species. Although birds and mosquitoes are the logical choice for surveillance programs on USUV circulation, horses could be included.

### 4.4. Donkey

WNND has been reported in donkeys in Brazil [65] and WNV neutralizing antibodies have been found in healthy donkeys and mules in Spain [66]. The Spanish study, however, did not describe having tested antibodies against flaviviruses other than WNV in the neutralization assay to confirm their IgG ELISA-positive results. The prevalence of TBEV antibodies among the 14 donkeys sampled in our study was remarkably high (4/14, 28.6%) and distributed over two federal states. The number of donkeys included in our study was, however, small and in addition largely gathered from one location, thus prohibiting generalization. There is very little published about donkeys and TBEV or USUV. A study among zoo animals in the Czech Republic found no neutralizing TBEV antibodies in the three sampled donkeys [67] and neither did a study including 201 donkeys in Pakistan [68], which is not surprising because TBEV is not endemic in Pakistan.

### 4.5. Risk Factors

The authors did not find explanatory risk factors for WNV nor TBEV seropositivity in the regression analysis. Risk factors reported in the literature for WNV seropositivity in horses are specific geographic areas (for example, along a major migratory bird route) [37,69,70], age [37,70], and breed (Quarter horse and Arabian) [36,66]. Interestingly, the breeds with the lowest risk were Warmbloods and ponies, which represent the predominant breeds in our study population. Breed susceptibility could reflect mosquito biting preference related to breed differences such as sweat composition or hair density [36]. Age has been found as a risk factor in different ways. Age seems a logical risk factor for seroprevalence in endemic areas, since the older a horse or human becomes, the more time it has had to be exposed to mosquitoes and thus to the virus. Indeed, a significantly lower WNV seroprevalence has been observed among the youngest age group of horses (1–3 yr) [37]. In humans, increasing age was associated with increasing seroprevalence [71]. In horses, age was found as risk factor for fatality as the outcome of clinical disease [72]. Further risk factors associated with WNV neutralizing antibodies were the number of horses within the holding, transport of the horse within the last six months, and the presence of mosquitoes in the holding [73]. Participating in polo was another risk factor for ELISA seropositivity in a multiple logistic regression model, but this could be confounding due to the geographic location of the polo stables in a hot and humid area [69]. Neutralizing antibody titers were not tested in that study. Nonetheless, the sports discipline of a horse could be related or confounded to management, breed, and traveling. Mosquito-control measures decreased the risk of WNV seropositivity. Some contradicting results regarding coat color were published with light colored horses having higher odds of fatal disease [74] and higher seroprevalence [37], in comparison to dark horses that more commonly display clinical signs [75]. For clear explanation of these findings, further seroprevalence investigations are needed.

## 5. Conclusions

This study substantiates WNV and TBEV endemicity in horses in eastern Austria, but could not unequivocally demonstrate USUV infections in equids. Since humans and equids as dead-end hosts share epidemiologic aspects, surveillance studies in equids like this are relevant for public health. Continuous surveillance of horses may be used as a predictor for virus circulation, which would allow medical and veterinary authorities to take action before large-scale infection of humans occurs.

## Figures and Tables

**Figure 1 viruses-13-01873-f001:**
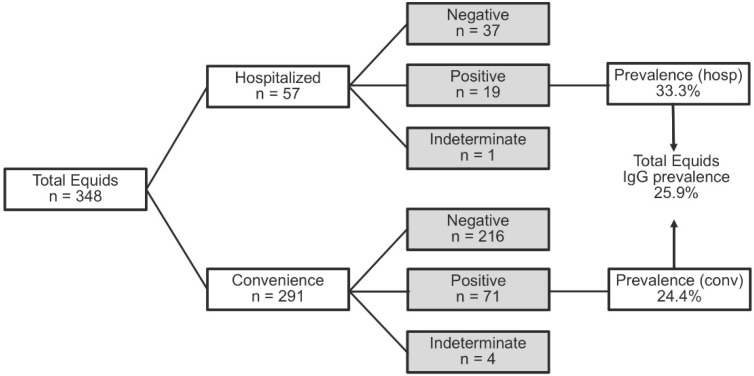
Laboratory results of IgG ELISA detecting flavivirus antibodies in serum. Abbreviation (hosp) refers to hospitalized equids and (conv) to convenience field sampled equids.

**Figure 2 viruses-13-01873-f002:**
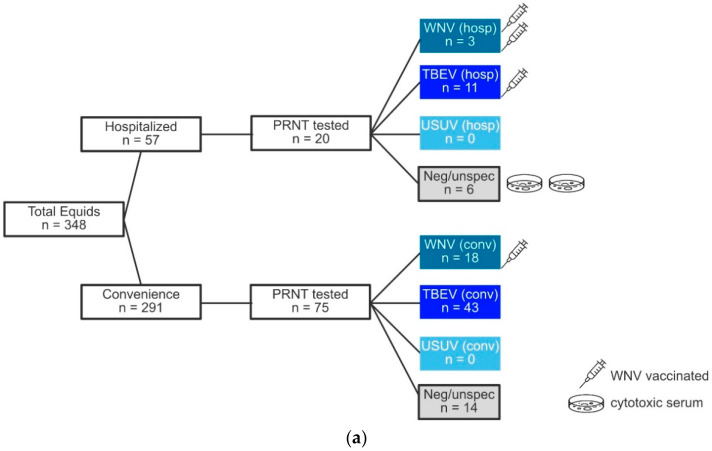
Laboratory results of PRNT-80: (**a**) IgG ELISA reactive or indeterminate samples tested with PRNT-80. Equids with cytotoxic serum, depicted by the symbol of a Petri dish, and WNV-vaccinated equids, depicted by the symbol of a syringe, are included. The number of symbols represents the number of equids; (**b**) Results of PRNT-80-positive samples are highlighted, negative samples are omitted. In two hospitalized horses, cytotoxic serum prohibited reliable neutralizing antibody analysis and they were therefore excluded, decreasing PRNT-80 group hosp to *n* = 55 and total equids (hosp + conv) to *n* = 346. Further subtracting WNV-vaccinated equids decreased equid numbers in hosp to *n* = 52, in conv to *n* = 290, and in total equids to *n* = 342. Abbreviations: PRNT = 80% plaque reduction neutralization test, WNV = West Nile virus, TBEV = Tick-borne encephalitis virus, USUV = Usutu virus, hosp = hospitalized equids, conv = convenience field sampled equids.

**Figure 3 viruses-13-01873-f003:**
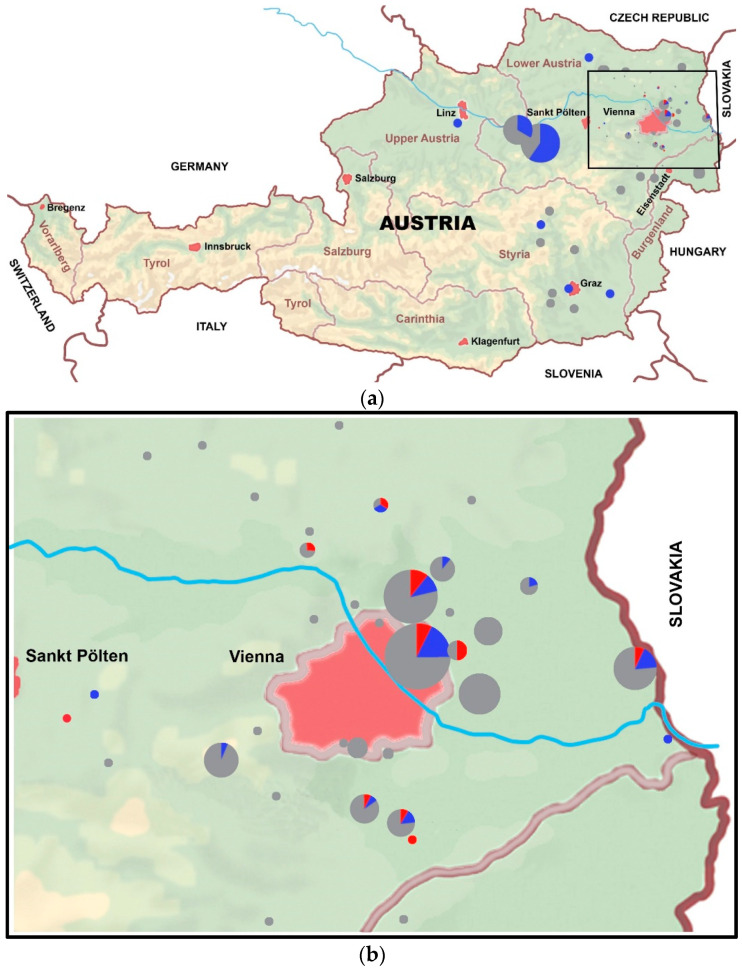
Maps of Austria showing the geographical locations using Google-derived GPS coordinates of the sampled equids (Google LLC, Mountain View, CA, USA). The size of the circles correlates with the number of equids sampled. Every equid is represented by the outcome of the highest reciprocal neutralizing antibody titer ≥ 20 (PRNT80). The colors represent neutralizing antibody status. Red represents WNV seropositive, blue TBEV seropositive, and grey flavivirus seronegative equids: (**a**) A map of Austria and neighboring countries. The inset highlights the greater Vienna area; (**b**) Close-up of Figure 3a (inset), highlighting the greater Vienna area.

**Table 1 viruses-13-01873-t001:** Overview of the study population: hospitalized (hosp; total 57) and convenience field sampled (conv; total, 291) equids distributed by breed and gender (hosp and conv) as well as reason for admission (hosp).

Population		Group Hosp Number (*n*)	Group Hosp Percentage (%)	Group Conv Number (*n*)	Group Conv Percentage (%)
**Breed**	Arabian and cross	1	1.8	13	4.5
	Donkey	1	1.8	13	4.5
	Haflinger and cross	9	15.8	12	4.1
	Icelandic horse	4	7.0	5	1.7
	Lipizzan	0	0	12	4.1
	Noriker	3	5.3	7	2.4
	Pony	1	1.8	28	9.6
	Quarter horse	5	8.8	18	6.2
	(Mini) Shetland pony	1	1.8	14	4.8
	Standardbred	3	5.3	15	5.2
	Thoroughbred and cross	0	0	4	1.4
	Warmblood	22	38.6	130	44.7
	Welsh pony and cross	0	0	4	1.4
	Other	7	12.3	11	3.8
	Unknown	0	0	5	1.7
**Gender**	Mare	26	45.6	117	40.2
	Gelding	30	52.6	149	51.2
	Stallion	1	1.8	24	8.2
	Not recorded	0	0	1	0.3
**Reason for hospital admission**	Orthopedic	21	36.8		
	Gastrointestinal	11	19.3		
	Dental	6	10.5		
	Dermatologic	4	7.0		
	Ophthalmologic	4	7.0		
	Respiratory	4	7.0		
	Fever ^1^	2	3.5		
	Urinary	2	3.5		
	Neurologic	2	3.5		
	Companion animal	1	1.8		

^1^ Horses with a singular complaint of fever are presented. Three other horses that presented with fever combined with gastrointestinal (*n* = 2) or orthopedic (*n* = 1) problems are categorized under their respective organ system.

**Table 2 viruses-13-01873-t002:** Relative risk ratios for selected variables. Variables with values greater than one indicate positive association with WNV or TBEV infections, respectively. For instance, for imported horses the relative risk for WNV infection increases by a factor of 2.55. However, all variables were non-significant.

Variable	Relative Risk Ratio Probable Agent	
	WNV	TBEV
Import (Yes vs. No)	2.55	1.56
Insect protection ^1^ (Yes vs. No)	0.56	1.56
Stable type (Outdoor vs. Box)	1.20	0.92
Coat color (Light vs. Dark)	0.56	1.47
Coat color (Twotone vs. Dark) ^2^	0.66	1.90
Illness ≤ 12 months	1.02	1.59
Stable federal state ^3^ (LA-east vs. Vienna) (LA-west vs. Vienna)	1.31 0.70	0.37 2.54

^1^ Insect protection included the use of mechanical (i.e., blankets) and/or chemical (i.e., repellent) insect control techniques. ^2^ A twotone coat color was defined as horses with a lot of white in the coat, not restricted to the head and legs; in this study, pinto and leopard colored horses. ^3^ For statistical analysis, the relatively large federal state of Lower Austria (LA) was analyzed as two states (LA-east and LA-west).

## Data Availability

Data are contained within the article or Appendix A. The data presented in this study are available in Appendix A.

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
