# Peer review of "West Nile Virus and Tick-Borne Encephalitis Virus Are Endemic in Equids in Eastern Austria"

_viruses, 2021, doi:10.3390/v13091873_

Round 1
Reviewer 1 Report
Overall, the study is presented in a clear and straightforward manner. The subject is of importance and is suitable for publication in Viruses. There are some points that I think the authors should address before the study is ready for publication. Mainly, the authors should address the problem of distinguishing between exposure to different Flaviviruses using the methods described in the study. Second, they should address the vectors issue, which is crucial for understanding the prevalence and for proper surveillance of the pathogens studied here.
There are other minor points, which I detailed below.
General comments:
- General comment regarding the core assay upon which the study is based - PRNT80 for Flaviviruses. The authors attempted to examine exposure to three species (WNV, TBE, USUV), which can be serologically similar in terms of seroconversion as reflected by the assay. Following the results, the seropositive animals were examined by PCR and were all negative.
If the assay is directed to detect IgG, it is not surprising, since the acute phase of infection was over and the chances of detecting the viral RNA in the blood are very low. In addition, even during acute infection, WNV viremia can be very low and barely detectable using PCR. I would expect that the authors would make some attempt to seek for a more specific serological assay that can distinguish (or at least provide a higher degree of confidence) between the pathogens. If no such assay was available at the time of the study, or if such assay was tested and was not successful, I think the authors should address it in the discussion. Serological differentiating between USUV and WNV is indeed an issue and since this is one of the main themes in this publication, I think it should be addressed more significantly, if not in the results, than at least in the discussion.
- Another general comment, which I mentioned in one of the specific points below, is the need to discuss information on the vectors – mosquitos for WNV and USUV, and ticks for TBE. Was this examined during the study? Are there previous reports on the vectors and their infection rate in the examined states / regions? I think this should be mentioned as well, to complete the conclusions of the study.
I attached the PDF with the specific comments embedded with the text.
Specific points:
Line 58 - Should be “acid WAS”. Since all the viruses included in this study are RNA viruses, the general term "nucleic acid" can be replaced by "RNA"
Line 171 - what are the units here? Which test was performed? The 80% plaque reduction assay? Other SNT?
Line 175 - I suggest that the authors consider using the word "marginal" or "borderline" instead of questionable, which may suggest that the result is not valid or not clear.
Line 207 - I am not sure I understood this sentence. All positive sera that had neutralizing Abs to USUV, were also active against another virus?
If not, please revise to clarify. If yes, was the specificity of the ELISA verified using standard controls? USUV and WNV can sometimes be indistinguishable serologically (depending on the kit used). Could it be that there was cross-reactivity of the antibodies with either WNV or TBEV?
Line 222 - I did not understand the sentence. The horses were raised in Hungary, or kept there? Alternatively, travelled through Hungary?
In addition, it is important to know in which part of the country. Was it close to Austria? Can the region of origin be considered a separate geographical area, compared with eastern Austria?
Figure 3 - the writing on the maps is unclear and the only word I could read was Lichtenstein. I suggest either improve the resolution of the maps, or use different maps that only show the state / district names and not towns and villages that cannot be read.
I think it is important to show the region where the stable was (or was assumed to be), and avoid including names and words that cannot be read.
Table 1. - are these subtitles or headings? Why are they separated from the other factors? I did not understand the arrangement of this table. Why not list all the variables instead of presenting them in a table?
If there is a rationale for arranging all variables in such a table, it was not clarified in the text.
Line 275 - it would have been surprising to find WNV RNA in horses with positive IgG. Unless horses with positive IgM were selected for the RNA test, one should not expect to find WNV or TBE RNA in horses that have a high IgG titer.
Line 295 - This is exactly what I was wondering - was there any correlation between the occurrence of WNV, as reflected by positive equid serology, and reports on WNF in humans or birds? Are there any data available from the districts that were examined?
I think this could add to the value of the study, to compare data from human serology.
In this respect, is there any information on mosquito surveillance in these regions? This could also add to the study.
Line 309 - I would also wonder how reliable and specific the ELISA kits are. Does the manufacturer mention the specificity of the assay - which species are expected to exert detectable reaction?
How likely is the ELISA to give non-specific signal?
Line 355 - as mentioned for the WNV, since TBE is a serious disease in humans, is there any information on TBE case in the regions where equid serology was positive, at the time of the study?
Line 377 - the same comment applies here - are there any data regarding human infections or tick vector studies that can complement the knowledge on TBE prevalence in the studied regions?
Line 388 - I would rephrase to "such as"
Line 408 - I would add that continuous surveillance in horses may be used as a predictor for virus circulation, and allow the authorities to take action before large-scale infection of humans occurs.
Supplementary table S1 – the table is very detailed, and there are some abbreviations that I did not understand or missed the corresponding explanation in the text. Some examples:
The abbreviations such as “h. 7xa, e. c. 2xa” in the Water sources column, the labelling in the Stabletype column, and the Transport where column – it is not clear whether this is the country of origin, or was the animal transported through that country, or something else.
I suggest that the authors either add a more detailed explanation regarding each column, or prepare a more concise table with less columns so that the reader can understand the details better.

Reviewer 2 Report
Overall a well written manuscript that highlights the importance of testing for arboviral pathogens.
Only minor comments:
In the abstract line 22-23, I would be helpful to the reader to include the year(s) that the samples were collected for the cross sectional study to relate it back to the previous sentence regarding the case detected in 2016.
In methods line 124, I suggest using "indeterminate" instead of "questionable".
Relating to RNA detection, there are several papers that highlight the importance of testing whole blood samples instead of serum for longer term detection of flaviviral RNA (up to 90 days in whole blood compared to ~5-7 days in serum). I would suggest performing this testing if those samples are available or including this in future studies.
Figure 3 is difficult to read with all the overlays from Google maps. Can you recreate this in another program?
Supplemental table 2 would be a beneficial table to include in the methods of the study, if space allows.
In the discussion, I would like to see more interpretation of what other studies may need to be done in the pathogen (especially about usutu) specific sections to help bring things together for the readers.
